# Paracrine Signals in Calcified Conditioned Media Elicited Differential Responses in Primary Aortic Vascular Smooth Muscle Cells and in Adventitial Fibroblasts

**DOI:** 10.3390/ijms24043599

**Published:** 2023-02-10

**Authors:** Amber M. Kennon, James A. Stewart

**Affiliations:** 1Department of Investigational Cancer, Division of Cancer Medicine, U.T.M.D Anderson Cancer Center, Houston, TX 77030, USA; 2Department of BioMolecular Sciences, School of Pharmacy, University of Mississippi, Oxford, MS 38677, USA

**Keywords:** vascular calcification, diabetes mellitus, vascular smooth muscle cells, adventitial fibroblasts, receptor for advanced glycation end-products (RAGEs), conditioned media

## Abstract

Our goal was to determine if paracrine signals from different aortic layers can impact other cell types in the diabetic microenvironment, specifically medial vascular smooth muscle cells (VSMCs) and adventitial fibroblasts (AFBs). The diabetic hyperglycemic aorta undergoes mineral dysregulation, causing cells to be more responsive to chemical messengers eliciting vascular calcification. Advanced glycation end-products (AGEs)/AGE receptors (RAGEs) signaling has been implicated in diabetes-mediated vascular calcification. To elucidate responses shared between cell types, pre-conditioned calcified media from diabetic and non-diabetic VSMCs and AFBs were collected to treat cultured murine diabetic, non-diabetic, diabetic RAGE knockout (RKO), and non-diabetic RKO VSMCs and AFBs. Calcium assays, western blots, and semi-quantitative cytokine/chemokine profile kits were used to determine signaling responses. VSMCs responded to non-diabetic more than diabetic AFB calcified pre-conditioned media. AFB calcification was not significantly altered when VSMC pre-conditioned media was used. No significant changes in VSMCs signaling markers due to treatments were reported; however, genotypic differences existed. Losses in AFB α-smooth muscle actin were observed with diabetic pre-conditioned VSMC media treatment. Superoxide dismutase-2 (SOD-2) increased with non-diabetic calcified + AGE pre-conditioned VSMC media, while same treatment decreased diabetic AFBs levels. Overall, non-diabetic and diabetic pre-conditioned media elicited different responses from VSMCs and AFBs.

## 1. Introduction

In a living system, cells communicate in an autocrine/paracrine fashion using molecular exchanges to stimulate responses from cells in their microenvironment. These localized soluble signals will enact changes in structures, such as the aorta. The aorta is composed of three layers or tunics: intima, media, and adventitia layers. The intima is mainly composed of endothelial cells lining the vessel lumen [1,2]. The medial layer contains vascular smooth muscle cells (VSMCs) responsible for vasocontraction and vasodilation of the vessel in response to stimuli [3]. The outermost layer, or the adventitia, is where adventitial fibroblasts (AFBs), extracellular matrix (ECM) components, progenitor cells, and immune cells reside [4]. In the physical space of the aorta, these cell layers experience the same microenvironment, meaning they are capable of sharing paracrine and autocrine signals to direct cellular processes in response to homeostatic and pathologic conditions.

In diabetes-mediated vascular calcification, cells encounter a hyperglycemic environment that potentiates a calcification response in the medial and adventitial layers of vessels. These factors can lead to vascular stiffness and loss of vascular compliance, which creates additional stress on the VSMCs and AFBs, leading to increased risk for cardiovascular complications [5]. Hyperglycemia also leads to the elevated presence of advanced glycation end-products (AGEs), which, when bound to their receptor, RAGE (receptor for AGE), activates downstream targets such as ERK1/2, P38 MAPK, NF-κB, and markers of oxidative stress (SOD-1 and SOD-2) [6,7,8,9,10,11,12,13,14,15]. In addition to signaling targets, RAGE can also modulate phenotype changes for VSMCs and AFBs such as α-smooth muscle actin (α-SMA), osteopontin (OPN), and vimentin [16,17,18,19]. Under diabetic calcification conditions, RAGE signaling plays a role in exacerbating the disease state by inducing the cells within the vessel to release paracrine or autocrine factors to modulate their surroundings.

Paracrine and autocrine factors collected in conditioned media, collected from in vitro modeling systems containing excreted factors, have been used to test cell–cell interactions [20,21,22]. Studies have shown AFBs treated with VSMC-conditioned media increased AFB migration [23]. Another study using hypoxic VSMC-conditioned media determined if angiogenesis would stimulate bovine VSMCs, AFBs, and endothelial cells to increase angiogenesis in all cell types [24]. Moreover, it is common for mesenchymal stem cell-conditioned media to be used to investigate its anti-inflammatory properties [25,26,27,28]. Using conditioned media from both VSMCs and AFBs, we sought to determine if factors in conditioned media from opposing cell types in an in vitro diabetic calcification model can elicit differential responses to exacerbate calcification in an AGE/RAGE dependent manner.

## 2. Results

### 2.1. Conditioned Media from Calcified AFBs Elicited Differential Calcification Responses in VSMCs

Non-diabetic VSMC calcification was significantly increased due to treatment with calcified + AGEs diabetic AFB conditioned media (Figure 1). Non-diabetic VSMCs showed significant differences in comparison to both RKO cell types in the non-diabetic calcified plus AGEs treatment, diabetic calcified treatment, and diabetic calcified and AGE treatment. Both RKO cell types generated no responses to any calcification treatment. Diabetic VSMCs treated with diabetic calcified plus AGEs-conditioned media demonstrated a significantly increased calcification response when compared to RKO cell types. Overall, non-diabetic VSMCs demonstrated the greatest response to diabetic calcified AFBs-conditioned media.

### 2.2. Conditioned Media from Diabetic AFBs Caused an Increase in OPN Expression in Diabetic RKO VSMCs

α-SMA expression was unchanged due to treatments, although there were significant genotypic differences (Figure 2A). OPN expression was significantly increased due to diabetic AFBs-conditioned media treatment (Figure 2B). This occurred specifically in the non-calcified + AGEs treatment group.

### 2.3. TLR4 and p38 MAPK Expressions Were Significantly Changed in RKO VSMCs

A lack of RAGE led to significantly increased TLR4 expression when compared to cells with RAGE (Figure 3A). There were no treatment differences in TLR4. P38 MAPK activation decreased between non-diabetic calcified conditioned media and diabetic calcified conditioned media in diabetic RKO cells (Figure 3B). Moreover, there were genotypic differences in p38 MAPK activation.

### 2.4. Conditioned Media from Diabetic AFBs Increased NF-κB Activation and SOD-2 Expression

NF-κB activation was not changed due to treatment in all genotypes except for diabetic cells (Figure 4A). Diabetic AFBs-conditioned media elicited a significant increase in NF-κB activation in diabetic VSMCs, especially in the calcified treatment group (Figure 4A). There were significant differences between genotypes in NF-κB activation and SOD-2 expression. Moreover, SOD-2 expression significantly increased in diabetic RKO VSMCs due to calcified plus AGEs diabetic AFBs-conditioned media (Figure 4B). This trend was also observed in the non-diabetic AFBs-conditioned media treated diabetic RKO cells. Generally, diabetic AFBs-conditioned media produced significant effects in diabetic and diabetic RKO VSMCs in NF-κB and SOD-2, respectively.

### 2.5. Messenger Differences Were Observed between Non-Diabetic and Diabetic AFBs-conditioned media

Many different chemokines, cytokines, and proteins were expressed within the AFBs-conditioned media collected and then applied to VSMCs (Figure 5). In all tested conditioned media samples, the relative expression of chemokine (C-X-C motif) ligand 1 (CXCL1), insulin-like growth factor binding protein 6 (IGFBP6), and matrix metalloproteinase-3 (MMP-3) were relatively high compared other factors (Figure 5). Non-diabetic-conditioned media samples expressed angiopoietin-2, growth/differentiation factor-15 (GDF-15), tissue factor, and WNT1-inducible signaling pathway protein 1 (WISP-1), while diabetic-conditioned media did not. Diabetic-conditioned media expressed Chemokine (C-C motif) ligand 6 (CCL6) and hepatocyte growth factor (HGF), while non-diabetic-conditioned media did not. The addition of AGEs to the conditioned media influenced the increased expression of chemokine (C-C motif) ligand 12/monocyte chemotactic protein-5 (CCL12/MCP-5) and chemokine (C-C motif) ligand 5 (CCL5). On the other hand, the addition or lack of AGEs influenced the decreased expression of insulin-like growth factor binding protein 2 (IGFBP2).

### 2.6. VSMCs-conditioned media Did Not Elicit a Differential Calcification Response in AFBs

Calcified diabetic-conditioned media showed some increases in calcification in diabetic cell types, but these changes were not significant (Figure 6). All treatments and genotypes were statistically insignificant.

### 2.7. Phenotypic Changes Were Observed in Diabetic and Non-Diabetic AFBs Due to VSMC-conditioned media Treatment

The application of calcified diabetic VSMC-conditioned media caused a decrease in α-SMA expression in non-diabetic AFBs (Figure 7A). Diabetic RKO AFBs α-SMA expression significantly decreased with the application of non-diabetic calcified plus AGEs-conditioned media when compared to the non-calcified plus AGEs-conditioned media. This trend was also present in the diabetic-conditioned media treated diabetic RKO AFBs, but insignificant (Figure 7A). Vimentin expression significantly decreased due to non-diabetic calcified plus AGEs-conditioned media treatment when compared to non-calcified treatments in diabetic AFBs (Figure 7B). Significant decreases in vimentin occurred between non-diabetic non-calcified groups and diabetic non-calcified groups in diabetic AFBs. Vimentin expression significantly increased due to treatment with diabetic calcified plus AGEs VSMC-conditioned media (Figure 7B). Overall, the interaction between treatment and genotype was significant for vimentin. Finally, OPN expression increased in diabetic AFBs due to diabetic non-calcified plus AGEs, calcified, and calcified plus AGEs-conditioned media treatment when compared to the non-calcified alternative (Figure 7C). This increase was also found in comparison to the non-diabetic calcified plus AGEs treatment group. The interaction of treatment and genotype was significant for OPN expression (Figure 7C). Finally, diabetic AFBs responded to VSMC-conditioned media treatment with changes in vimentin and OPN expression.

### 2.8. VSMC-conditioned media Influenced P38 MAPK Activation

Treatment with conditioned media did not cause changes in TLR4 expression, but there were genotypic differences, such as diabetic cells expressing less TLR4 than other genotypes (Figure 8A). There were no changes in ERK 1/2 activation due to treatment or genotype (Figure 8B). In diabetic cells, P38 MAPK activation significantly decreased when comparing non-diabetic calcified groups to diabetic non-calcified groups (Figure 8C). Moreover, P38 MAPK activation decreased in diabetic cells when comparing non-diabetic to diabetic-conditioned media in the non-calcified plus AGEs treatment group. There were significant genotypic differences observed in P38 MAPK activation (Figure 8C). P38 MAPK activation was the only signaling factor that gave detectable differences due to treatment.

### 2.9. AFB Non-Diabetic Cell SOD-2 Expression Changed Due to Treatment withVSMC Conditioned Media

NF-κB activation and SOD-1 expression did not change due to treatment with conditioned media although there were significant genotypic differences (Figure 9A,B). Non-diabetic and non-diabetic RKO AFBs showed significantly increased SOD-2 expression with application of non-diabetic calcified plus AGEs-conditioned media (Figure 9C). Diabetic calcified plus AGEs conditioned influenced a reduction in SOD-2 expression in non-diabetic cells when compared to non-diabetic calcified and AGEs-conditioned media. Non-diabetic RKO AFBs had significantly decreased SOD-2 due to the application of diabetic calcified conditioned media. Overall, SOD-2 expression showed significant genotype and treatment differences (Figure 9C).

### 2.10. Messenger Differences Were Observed in Calcified VSMCs-conditioned media Depending upon Genotype and AGE Presence

All tested VSMC-conditioned media samples produced cystatin C, IGFBP2, IGFBP6, MMP-3, and OPG at a relative value greater than 0.7 (Figure 10). CXCL1, CXCL16, and interleukin-6 (IL-6) were differentially expressed in the presence of AGEs, while tissue factor was only expressed without the presence of AGEs. Interestingly, CCL20 was only expressed in the diabetic calcified plus AGEs-conditioned media.

## 3. Discussion

The goal of this work is to begin to understand the role of paracrine and autocrine factors collected in conditioned media from our in vitro modeling systems to test cell–cell interactions occurring between VSMCs and AFBs in the diabetic aorta. Using conditioned media from both VSMCs and AFBs exposed to calcification and exogenous AGE treatment, we sought to determine if secreted factors in conditioned media from opposing cell types can elicit differential protein signals to exacerbate calcification in an AGE/RAGE dependent manner.

### 3.1. Conditioned AFB Media Applied to VSMCs

While few significant protein expression changes occurred in VSMCs because of AFB CM exposure, these cells responded with an overall increase in their calcification response. For example, non-diabetic VSMCs had a higher amount of calcification due to exposure to calcified AFB CM with AGEs. In fact, all non-diabetic VSMCs had an increased calcification response due to exposure to AFB CM from non-diabetic calcified plus AGEs, diabetic calcified, and diabetic calcified plus AGEs. In addition, diabetic VSMCs calcified in response to the same AFB CM groups. Alternatively, diabetic RKO VSMCs did not yield a calcification response. While changes in protein expression were limited, there were key osteogenic and inflammatory/oxidative stress markers that were elevated to indicate CM from AFBs played a role in mediating a paracrine calcification response. For example, diabetic RKO VSMCs produced significant protein changes in OPN and P38 MAPK when exposed to CM from diabetic, non-calcified plus AGEs. CM from non-diabetic, non-calcified plus AGE treated AFBs resulted in an increased diabetic RKO VSMC OPN expression. Interestingly, diabetic calcified AFB CM decreased P38 MAPK expression in diabetic RKO VSMCs in comparison to non-diabetic calcified AFB CM. Diabetic VSMC NF-κB activation was significantly increased due to exposure to calcified diabetic AFB CM when compared to exposures to non-diabetic calcified AFB CM. Finally, diabetic RKO VSMC SOD-2 expression significantly increased due to diabetic calcified plus AGEs AFB CM compared to all other groups. While VSMCs produced some significant protein changes in response to exposure to different AFB CMs, the overall response was a push towards a proinflammatory-oxidative stress pathway in VSMCs which also appeared to be RAGE-dependent.

The increases in calcification could be due to exogenous AGEs in the conditioned media or the presence of chemokines or cytokines in the AFB CM. All three AFB CM groups expressed lipopolysaccharide-induced CXC chemokine (LIX; CXCL5), which has been found in greater concentrations in the cardiovascular system than any other organ systems in the body [29]. LIX/CXCL5 are released from fibroblasts to act as a neutrophil attractant [29]. High levels of LIX/CXCL5 have been associated with coronary artery disease and postulated to be a potential biomarker and pharmacological target for this pathology [30]. Therefore, the expression of LIX/CXCL5 might be contributing to the increased VSMC calcification due to calcified AFB CM treatment. Studies have also shown that inorganic phosphate and AGEs will induce calcification in VSMCs; however, for our studies, diluted CM was utilized, meaning AGEs and calcification media concentrations were diluted prior to exposure [14,31]. Observed changes may be slightly dampened due to dilution of chemokine presence.

With diabetic RKO VSMCs, OPN expression was significantly increased when diabetic, non-calcified plus AGEs AFB CM was used as compared to non-diabetic non-calcified plus AGEs AFB CM. The presence of high glucose levels from the diabetic AFB CM may have caused increased OPN expression. Researchers have shown that OPN was increased in VSMCs treated with high glucose [32,33]. While there were minimal phenotype changes in any cell type, we further investigated signaling proteins shown to impact the AGE/RAGE cascade. TLR4 expression was significantly higher in RKO VSMCs. The lack of RAGE expression may cause TLR4 to be upregulated to compensate for AGE-related changes as both TLR4 and RAGE share adaptor and signaling proteins capable of activating some of the same signaling pathways [12,34]. In addition to TLR4, the activation of p38 MAPK in diabetic RKO VSMCs was revealed to have a decreased phosphorylation state due to diabetic calcified AFB CM compared to non-diabetic calcified AFB CM. This finding may result from the increased presence of C-X3-C motif chemokine ligand 1 (CX3CL1) in the non-diabetic calcified AFB CM compared to the diabetic calcified AFB CM. CX3CL1 has been shown to signal through P38 MAPK in an Alzheimer’s disease study [35]. Therefore, CX3CL1 may have influenced a higher P38 MAPK activation in the non-diabetic calcified AFB CM-treated diabetic RKO VSMCs than the diabetic calcified AFB CM-treated diabetic RKO VSMCs. In addition to changes in p38 MAPK, diabetic AFB CM elicited increased NF-κB activation in diabetic VSMCs. Interestingly, the diabetic AFB CM expressed hepatocyte growth factor (HGF) while non-diabetic AFB CM did not. HGF has been demonstrated to signal through NF-κB, which may have resulted in our finding of increased NF-κB activation [36]. In addition to diabetic NF-κB changes, SOD-2 expression increased due to calcification plus AGEs diabetic AFB CM treatment in diabetic RKO VSMCs. Calcified plus AGEs diabetic AFB CM expressed higher amounts of C-C motif chemokine ligand 20 (CCL20) than all other AFB CM groups tested. SOD-2 expression was linked to an increase in reactive oxygen species, which causes the release of CCL20 to recruit lymphocytes [37]. The presence of CCL20 indicated reactive oxygen species—leading to a feed-forward loop resulting in increased SOD-2 expression—which confirms our findings [37]. Although other genotypes did not respond with increased SOD-2 expression, diabetic RKO cells could have shown a response due to the lack of RAGE and diabetic phenotype, which have both been linked to SOD-2 regulation [38,39]. The applied AFB CM elicited different protein responses, but there were some interesting differences in the AFB CM contents not already outlined above.

Briefly, CXCL1, IGFBP6, and MMP-3 were found at relatively high levels in all samples. Senescent fibroblasts produce CXCL1, which acts as a chemoattractant for neutrophils [40,41]. IGBP6 has also been linked to senescence in fibroblasts, and it regulates VSCMs proliferation in varicose veins [42,43]. MMP-3 (matrix metalloprotease-3 or stromelysin-1) has been demonstrated to be responsible for the breakdown of ECM proteins and activation of other MMPs, such as MMP-1, MMP-7, MMP-9 [44]. MMP-3 and MMP-7 have been shown to cleave differing portions of OPN [44]. Angiopoietin-2, GDF-15, tissue factor and WISP-1 were also differentially expressed in non-diabetic AFB CM over diabetic AFB CM. Angiopoietin-2 signaling is linked to the induction of inflammatory responses in the endothelial cells of the vessel [45,46]. GDF-15 is also associated with an inflammatory response and mortality in T2DM [47,48]. Tissue factor has been demonstrated to be involved in the clotting response due to injury or stress [49]. WISP-1 regulates mitosis and survival in fibroblasts and has been associated with response to ischemic stress [50,51]. The presence of CCL12/MCP-5 and CCL5 was exogenous AGE dependent, and both have been associated with RAGE signaling [52,53]. Finally, IGFBP2 expression was decreased in groups not treated with exogenous AGEs, which has been found in rat osteosarcoma cells [54]. Overall, AFBs CM produced an exciting array of cytokines, chemokines, proteins, and growth factors which would benefit from further investigation.

### 3.2. VSMC-Conditioned Media Applied to AFBs

AFBs did not have a strong calcification response with exposure to VSMC CM addition, but there were marked protein expression changes observed in exposed AFBs. Diabetic AFBs significantly increased OPN and vimentin expression due to calcified plus AGEs VSMC CM treatment. At the same time, non-diabetic AFBs showed decreased α-SMA expression due to calcified plus AGEs VSMC CM treatment. VSMC CM treatment did not cause changes in TLR4 and ERK 1/2, but diabetic AFBs responded to non-calcified plus AGEs diabetic VSMC CM with decreased P38 MAPK activation. Finally, all non-diabetic AFBs increased SOD-2 expression due to calcified plus AGEs non-diabetic VSMC CM. Moreover, all non-diabetic AFBs showed decreased SOD-2 expression due to calcified plus AGEs diabetic VSMC CM treatment. Overall, AFBs produced some significant protein changes in response to VSMC CM.

AFBs did not produce a differential calcification response due to the addition of VSMC calcified CM from any genotype, even with the addition of exogenous AGEs. Interestingly, AFBs did have significant protein changes due to VSMC CM treatment. Non-diabetic AFBs significantly decreased α-SMA expression due to calcified plus AGEs diabetic VSMC CM treatment compared to non-calcified diabetic VSMC CM treatment. α-SMA is regarded as an ‘activated’ fibroblast marker, and decreases in α-SMA could have indicated the transition of AFBs to other cell types that may have either an osteogenic or a macrophage-like cell fate [19,55,56,57,58]. Moreover, decreased α-SMA in non-diabetic AFBs may have been influenced by the high levels of CCL20 found in the diabetic calcified plus AGEs VSMC CM. Calvayrac et al. found α-SMA was co-localized with CCL20 in human atherosclerotic plaques [59]. In addition, CCL20 has been implicated in the upregulation of vimentin in hepatocellular carcinoma, which supports our findings that vimentin was significantly increased in diabetic AFBs due to calcified plus AGEs diabetic VSMC CM treatment [60]. Moreover, increased vimentin in diabetic AFBs could be due to the elevated level of pentraxin-3 in calcified plus AGE, diabetic VSMC CM, which has been implicated in endothelial-mesenchymal transition [61,62]. Diabetic AFBs also experienced increased OPN expression due to diabetic VSMC CM treatment, which could be attributed to the presence of high glucose in the media similar to the findings observed in VSMCs [32,33,63]. Overall, diabetic AFBs may have begun to transition away from the fibroblast phenotype towards a different cell type in response to diabetic VSMC CM treatment.

Due to changes in AFBs phenotype markers, we wanted to investigate further signaling markers related to RAGE signaling. TLR4 and ERK1/2 showed no significant changes due to CM treatment. At the same time, P38 MAPK activation decreased in diabetic AFBs due to non-calcified plus AGEs diabetic VSMC CM compared to non-calcified plus AGEs non-diabetic VSMC CM. This change may be due to the diabetic cell type or the diabetic VSMC CM, but more work will need to be performed to elucidate further. NF-κB activation was investigated, but no significant changes due to treatment were found. Moreover, SOD-1 expression was unchanged. Interestingly, SOD-2 expression increased in non-diabetic AFBs due to non-diabetic calcified plus AGEs VSMC CM treatment. High levels of IL-6 have been associated with increased SOD-2 expression, but IL-6 was also present in the diabetic calcified plus AGEs VSMC CM, which was not observed to have increased SOD-2 expression [64]. Although all differences could not be fully explained, this study has provided unique data to suggest that AFBs can respond to VSMC CM.

Array analysis of VSMC CM demonstrated the presence of different array cytokines and chemokines. IGFBP6 and MMP-3 were also expressed in all VSMC CM samples, which were discussed above. Briefly, IGBP6 can cause senescence in fibroblasts, and MMP-3 cleaves OPN [42,43,44]. Cystatin C, IGFBP2, and osteoprotegerin (OPG) were also expressed in all VSMC CM samples. Cystatin C has been used as a biomarker for cardiovascular risk associated with chronic kidney disease [65]. Moreover, OPG has been found in the arterial walls of diabetic patients and is utilized as a bone marker for VSMC [20,66]. IGFBP2 release can become upregulated in VSMCs due to serum deprivation, which may be possible be as a result of using low serum media in our VSMC studies [67]. The presence of AGEs in the VSMC CM influenced the expression of CXCL1, which has been demonstrated to be connected to RAGE in airway inflammation [68]. Moreover, CXCL16 was upregulated in the presence of AGEs in human macrophages [69]. Interestingly, we found that tissue factor was decreased in the sample with AGEs. Published findings have shown that AGEs upregulate tissue factor; however, more research is required to elucidate this further [70,71,72].

In summary, we found increased SOD-2 expression in both cell types due to treatment, which could implicate elevated ROS in our findings. Prospective studies would include measurements of ROS, VSMCs responded to AFBs CM with calcification and protein changes, but AFBs did not react to VSMCs CM with calcification changes. AFBs did have some protein expression changes; however, our findings lead us to hypothesize that AFBs may be responsible for in part mediating or exacerbating VSMC calcification. Future studies are required to determine how these two cell types influence each other in vivo.

## 4. Materials and Methods

### 4.1. Animal Model

Genetically diabetic male mice (BKS.Cg-*Dock7^m^+/+Lepr^db^*/J; Jackson Labs; JAX# 00642; Bar Harbor, MN, USA) were used as the animal model for this study. The db/db mouse lacks the leptin receptor due to a point mutation, which causes polyphagia leading to hyperglycemia and eventually type 2 diabetes. Littermate controls (heterozygous, db/wt, non-diabetic) were used as the lean control group. These mice cannot be distinguished from wild-type mouse. The RAGE knockout model was generated as outlined in Burr et.al., Lee et. al., Liliensiek et. al., and Schwenk et. al [73,74,75,76]. The genotypes generated for this study are derived from a cross between the db/wt mouse and the RKO mouse. The groups are as follows: non-diabetic (db/wt, non-db), diabetic (db/db, db), non-diabetic RAGE knockout (db/wt^RKO^, non-db RKO), and diabetic RAGE knockout (db/db^RKO^, db RKO). All mice had a C57BL6 background and were in an overt diabetic state marked by hyperglycemia when taken at 16 weeks of age [77]. Each isolation contained aortic layers from 3–4 mice and data from 3–6 separate isolations were collected per genotype. All mice were group-housed in an AAALAC-approved animal facility following the National Institutes of Health “Guide for the Care and Use of Laboratory Animals.” Mice experienced a 12 h/12 h light/dark cycle, and food and water were ad libitum. The University of Mississippi Animal Care and Use Committee (IACUC protocol number 20-017) approved all animal usage protocols.

### 4.2. Primary Murine Vascular Smooth Muscle Cell (VSMCs) Isolation and Culture

CO_2_ asphyxiation, followed by cervical dislocation, were used to euthanized animals [78,79]. Body weight and non-fasting blood glucose measurements were taken at the time of euthanasia, followed by removal of the thoracic aorta (Appendix A; https://doi.org/10.6084/m9.figshare.22062644) [80]. Briefly, the adventitial layer was physically separated from the medial layer. The adventitial layer was used for AFBs isolations detailed later in the Section 4. The medial layer was digested in a collagenase elastase solution [1950 U/A of collagenase type 2 enzyme (Worthington Biochemical; Lakewood, NJ, USA), 11.275 U/A elastase enzyme (Worthington Biochemical; Lakewood, NJ, USA), 0.004% Trypsin (Corning; Durham, NC, USA), and 10 mL of High-Glucose Dulbecco’s Modified Eagles Medium (HG-DMEM, 4.5 g/L glucose; Corning; Durham, NC, USA)] for 45 min at 37 °C under constant agitation. The mixture was strained through a cell strainer, centrifuged at 225× *g* for 10 min, and resuspended in VSMC HG-DMEM [14.3 mM NaHCO3, 15 mM HEPES, 15% FBS, 2% L-glutamine (Corning; Durham, NC, USA), 2X Primocin™ (Invivogen; Carlsbad, CA, USA), and Clonetics® Smooth Muscle Growth Media-2 SingleQuots (Lonza, Houston, TX, USA)]. Remaining tissue was placed in a collagenase digestion solution [100 U/mL type two collagenase, 0.1% trypsin (Gibco ThermoFisher Scientific, Waltham, MA, USA), and HG-DMEM] for 30 min under constant agitation at 37 °C until all tissue was digested. All solutions were combined and centrifuged 225× *g* for 10 min. The pellet was resuspended, and cells were plated on 100 μg/mL PureCol® collagen-coated (Advanced Biomatrix; Carlsbad, CA, USA) 60 mm plates in VSMC HG-DMEM. Cells were washed with appropriate glucose media according to genotype (i.e., HG-DMEM for diabetic and low glucose-DMEM (LG-DMEM, 1 g/L glucose- euglycemic media; Corning; Durham, NC, USA) for non-diabetic) 24 h after plating. VSMCs were utilized at P1 for baseline characterization and P2 for signaling experiments. Each isolation contained aortic medial layers from 3–4 mice and data from 3–6 separate isolations were collected per genotype.

### 4.3. Primary Murine Adventitial Fibroblast (AFBs) Isolation and Culture

As stated previously, CO_2_ asphyxiation, followed by cervical dislocation, were used to euthanize animals [78,79]. Body weight and non-fasting blood glucose measurements were taken at the time of euthanasia, followed by removal of the thoracic aorta (Appendix A; https://doi.org/10.6084/m9.figshare.22062644) [80]. Briefly, the adventitial layer was physically separated from the medial layer. The adventitial layer was digested in 10-min increments under constant agitation at 37 °C in collagenase digestion solution described above. Each round of digestion was combined with the previous harvest. After cell isolate suspension was centrifuged at 225× *g* for 10 min, the cell pellet was resuspended in AFB HG-DMEM (14.3 mM NaHCO3, 15 mM HEPES, 15% FBS, 2% L-glutamine, and 2X Primocin™) cell solution. After all tissue was digested, cell suspensions were combined and centrifuged at 225× *g* for 10 min. The pellet was resuspended, and cells were plated in AFB HG-DMEM. Cells were washed with appropriate glucose media according to genotype (i.e., HG-DMEM for diabetic and low glucose-DMEM (LG-DMEM, 1 g/L glucose- euglycemic media; Corning; Durham, NC, USA) for non-diabetic) 24 h after plating. AFBs were utilized at P1 for baseline characterization and P2 for signaling experiments. Each isolation contained aortic adventitia from 3–4 mice and data from 3–6 separate isolations were collected per genotype.

### 4.4. Conditioned Media Collection and Application

Conditioned media were collected from non-diabetic and diabetic VSMC and AFB in vitro calcification experiments on day 7. Briefly, cells were passaged until P2 and then washed in 1x sterile PBS. Cell division was reduced with 2% FBS genotype appropriate DMEM to conduct calcification experiments. Relevant groups were treated with 0.5 mg/mL albumin (glycated, human AGEs, Sigma-Aldrich; St. Louis, MO, USA) and/or 3 mM inorganic phosphate at the start of each study and reapplied on day three. Calcification studies were concluded at day 7 and conditioned media was collected at that time. Therefore, the media was conditioned for four days in all four treatment conditions (No Calcification, No Calcification + AGE, Calcification, Calcification + AGE). Media was collected, centrifuged at 225× *g* for 10 min at 4 °C and the supernatant was placed into a new tube. For experiments, a pooled collection of 3–4 *n* values was combined for application to cells. Cells to be treated were rinsed in 1x sterile PBS. The conditioned media was applied at a dilution of 1:10 in 2% FBS genotype appropriate DMEM onto cells and changed at day 3 (Appendix A; https://doi.org/10.6084/m9.figshare.22062644). The conditioned media was reapplied on day 3, and the studies were carried out until day seven. Treatment groups are outlined in Appendix A (https://doi.org/10.6084/m9.figshare.22062644) and visually depicted in Appendix A (https://doi.org/10.6084/m9.figshare.22062644).

### 4.5. Calcification Quantification

Calcium quantification was performed as previously outlined [80]. Briefly, cells were seeded onto 96-well plates with matched wells for calcification and cell number. We utilized 60 mm plates for protein collection. Cells were washed in 1x PBS twice, and then 250 μL 0.6 N HCl was placed on the cells for 24 h. The HCl supernatant was collected after 24 h. The HCl supernatant’s calcium content (μg) was measured with the Calcium Colorimetric Kit (MAK-022; Sigma-Aldrich; St. Louis, MO, USA), following the manufacturer’s instructions.

### 4.6. DAPI Staining for Cell Number

Cells were fixed with 4% PFA on the calcification matched wells for 10 min and then washed with 1x PBS twice. 1:1000 DAPI in 0.01% Triton X-100 in 1x PBS was placed on the cells overnight at 4 °C. After incubation, cells were washed in 1x PBS twice and imaged with high content analysis Nikon Ti2-E microscope using a high-speed PCOS camera. Images obtained were analyzed with high content analysis software with automated cell counting, cell growth analysis, proximity analysis, and Nikon propriety JOBS analysis. Calcium content (µg) was normalized to cell number. Each experimental replicate is representative of 3 experimental repeats for each group.

### 4.7. Protein Analysis

Protein collection and western blotting was carried out as outlined in Kennon and Stewart [80]. Briefly, total protein was isolated with modified Hunter’s Buffer (1% Triton X-100, 75 mM NaCl, 5 mM Tris (pH 7.4), 0.5 mM orthovanadate, 0.5 mM EDTA, 0.5 mM EGTA, 0.25% NP-40 and Halt-Protease Inhibitor Cocktail (100X; 78430; ThermoFisher Scientific; Waltham, MA, USA)). Protein concentration for each sample was determined using the bicinchoninic acid (BCA) assay (23225; Pierce™ Biotechnology; ThermoFisher Scientific; Waltham, MA, USA) according to the manufacturer’s instructions. Proteins were separated on a 12% SDS-PAGE gel and transferred to methanol-activated Immobilon-P PVDF membrane (Millipore Sigma; ThermoFisher Scientific; Waltham, MA, USA). The membranes were blocked, and primary antibodies were placed on the membrane in either 5% milk or 5% BSA diluted in TBS-T (50 mM Tris-Base, 100 mM NaCl, and pH 7.4 with 0.001% Tween-20) overnight at 4 °C. The following primary antibodies were utilized: α-SMA (1:1000; A2547; Sigma-Aldrich; St. Louis, MO, USA), vimentin (1:1000; Cell Signaling, Danvers, MA, USA; 5741), osteopontin (OPN; 1:400; Abcam, ab8448), toll-like receptor 4 (TLR4; 1:400; Santa Cruz Biotechnology; Dallas, TX, USA;, sc-293072), phosphorylated ERK ½ (1:400; Santa Cruz Biotechnology; Dallas, TX, USA; sc-7383), total ERK 1 (1:400; Santa Cruz Biotechnology; Dallas, TX, USA; sc-271269), total ERK 2, (1:400; Santa Cruz Biotechnology; Dallas, TX, USA; sc-1647), phosphorylated NF-κB p65 (1:400; Santa Cruz Biotechnology; Dallas, TX, USA; sc-136548), total NF-κB p65 (1:400, Santa Cruz Biotechnology; Dallas, TX, USA; sc-8008), phosphorylated p38 MAPK (1:1000; Cell Signaling; Danvers, MA, USA; 9211), total p38 MAPK (1:1000; Cell Signaling; Danvers, MA, USA; 9212), superoxide dismutase 1 (SOD-1; 1:400; Santa Cruz Biotechnology, Dallas, TX, USA; sc-101523), superoxide dismutase 2 (SOD-2; 1:400; Santa Cruz Biotechnology; Dallas, TX, USA; sc-133134), and β-tubulin (1:400; Santa Cruz Biotechnology; Dallas, TX, USA; sc-398937) as a loading control. The membrane was washed with TBS-T and then incubated at room temperature with a secondary antibody conjugated to horseradish peroxidase (HRP) (Santa Cruz Biotechnology; Dallas, TX, USA;). The membrane was washed and then incubated in Pierce enhanced chemiluminescent substrates (ThermoFisher Scientific; Waltham, MA, USA) for 2 min. The membrane was visualized with the iBRIGHT western blot imaging system (ThermoFisher Scientific; Waltham, MA, USA), and densitometric analysis on bands was performed with NIH Image J software.

### 4.8. Proteome Profiler™ Array

The mouse XL cytokine array kit (ARY208; R&D Systems; Minneapolis, MN, USA) was utilized to analyze the contents of the conditioned media collected from calcified VSMCs and AFBs. Manufacturer’s instructions were followed for the duration and analysis of the kit. Briefly, membranes were blocked with array buffer 6, then 1 mL of array buffer 6 was added to 500 μL of pooled conditioned media and then to the membranes. Pooled conditioned media and membranes were incubated on a rocking shaker platform overnight at 2 °C. Membranes were washed in the kit-provided wash buffer and then incubated with the detection antibody cocktail for 1 h at room temperature on a rocking shaker platform. Membranes were washed again and then incubated with Streptavidin-HRP for 30 min at room temperature on a rocking shaker platform. Membranes were washed, removed from wash buffer, and placed on an iBRIGHT western blot imaging system (ThermoFisher Scientific; Waltham, MA, USA). The Chemi Reagent Mix was applied to each membrane, and then they were exposed and imaged for 10 min. For analysis, the average of duplicate spots was taken and then normalized to the average of all reference spots for the specific membrane. This generated a heat map that corresponds to relative changes in analyte between samples.

### 4.9. Statistical Analysis

Data were presented as mean ± standard error of the mean. A two-way ANOVA was performed with GraphPad Prism software, version 9.0.2 (Boston, MA, USA). All figures were analyzed with a Fisher’s LSD post hoc. This analysis method was performed because the analysis’s overall *p*-value must be significant for the post hoc to detect significance. Fisher’s LSD does not consider comparisons between other groups, and each comparison stands alone. Post hoc results on graphs are focused on groups of interest.

## Figures and Tables

**Figure 1 ijms-24-03599-f001:**
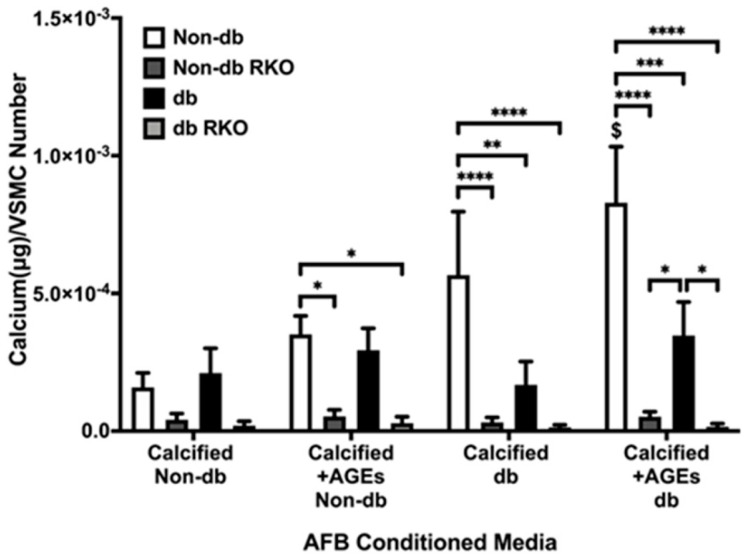
VSMCs responded to diabetic and non-diabetic AFB conditioned media. Primary VSMCs were isolated and treated with AFBs-conditioned media at a dilution of 1:10 in 2% FBS genotype appropriate DMEM for 7 days. Calcium (μg) was normalized to the number of cells (DAPI) and graphs as mean ± SEM with *n* = 3–6 of independent replicates. Statistical analysis consisted of a two-way ANOVA and a Fisher’s LSD test post hoc ($ means *p* < 0.05 when compared to non-db CM in the same genotype of treated cells, * *p* < 0.05, ** *p* < 0.01, *** *p* < 0.001, **** *p* < 0.0001).

**Figure 2 ijms-24-03599-f002:**
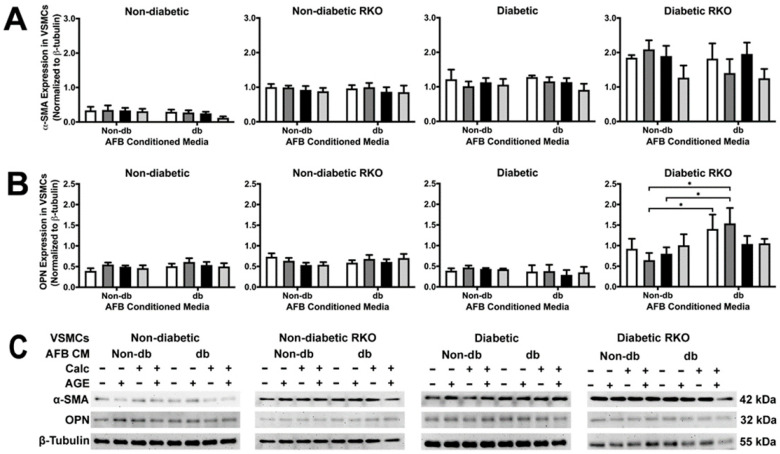
Diabetic RKO VSMCs had more α-SMA and OPN expression than other genotypes while diabetic RKO cells showed an increase in OPN due to diabetic AFBs-conditioned media application. Primary VSMCs were isolated and treated with AFBs-conditioned media at a 1:10 dilution in 2% FBS genotype appropriate DMEM for 7 days. Protein expression was quantified of α-SMA (42 kDa: (**A**)) and OPN (32 kDa: (**B**)). All protein levels were normalized to β-tubulin (**C**) and graphed as mean ± SEM with *n* = 3–6 of independent replicates. A two-way ANOVA with Fisher’s LSD post hoc was performed (* *p* < 0.05) on each graph individually. All data were analyzed together with a two-way ANOVA to give the following: α-SMA (Interaction; *p* = 0.7660, Genotype; *p* < 0.0001, and Treatment; *p* = 0.1193) and OPN ((Interaction; *p* = 0.7430, Genotype; *p* < 0.0001, and Treatment; *p* = 0.4289).

**Figure 3 ijms-24-03599-f003:**
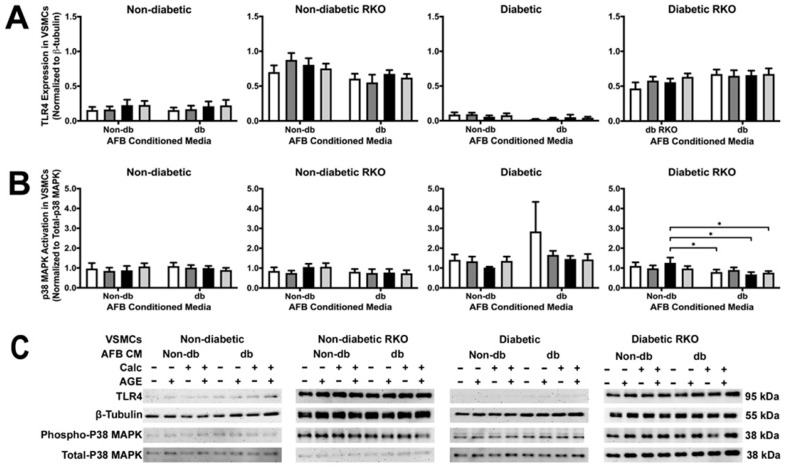
TLR4 expression was increased in RKO cell types, while p38 MAPK activation decreased due to calcified diabetic AFBs-conditioned media in diabetic RKO cells. Primary VSMCs were isolated and treated with AFBs-conditioned media at a 1:10 dilution in 2% FBS genotype appropriate DMEM for 7 days. Protein expression was quantified of TLR4 (95 kDa; (**A**)) and p38 MAPK (38 kDa, (**B**)). All protein levels were normalized to β-tubulin (**C**) and graphed as mean ± SEM with *n* = 3–6 of independent replicates. A two-way ANOVA with Fisher’s LSD post hoc was performed (* *p* < 0.05) on each graph individually. All data were analyzed together with a two-way ANOVA to give the following: TLR4 (Interaction; *p* = 0.5079, Genotype; *p* < 0.0001, and Treatment; *p* = 0.7425) and p38 MAPK (Interaction; *p* = 0.2778, Genotype; *p* < 0.0001, and Treatment; *p* = 0.0357).

**Figure 4 ijms-24-03599-f004:**
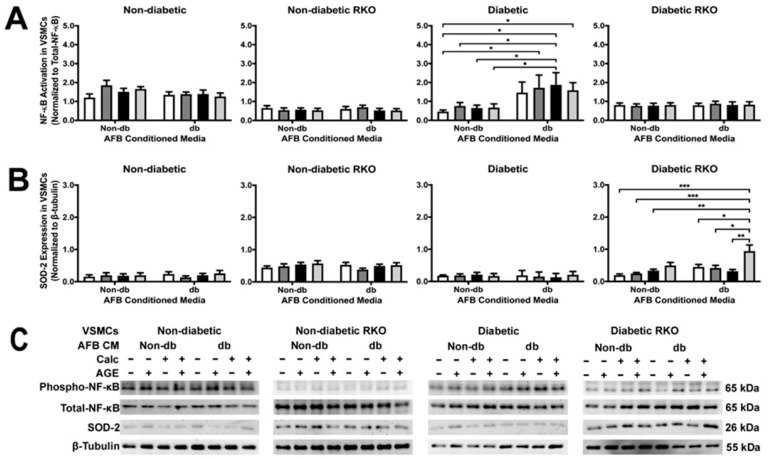
Diabetic AFBs-conditioned media increased NκB activation in diabetic VSMCs and increased SOD-2 expression in diabetic RKO VSMCs. Primary VSMCs were isolated and treated with AFBs-conditioned media at a 1:10 dilution in 2% FBS genotype appropriate DMEM for 7 days. Protein expression was quantified of p65 NF-κB (65 kDa: (**A**)) and SOD-2 (26 kDa: (**B**)). All protein levels were normalized to β-tubulin (**C**) and graphed as mean ± SEM with *n* = 3–6 of independent replicates. A two-way ANOVA with Fisher’s LSD post hoc was performed (* *p* < 0.05, ** *p* < 0.01, *** *p* < 0.001) on each graph individually. All data were analyzed together with a two-way ANOVA to give the following: NF-κB (Interaction; *p* = 0.0019, Genotype; *p* < 0.0001, and Treatment; *p* = 0.0267) and SOD-2 (Interaction; *p* = 0.0648, Genotype; *p* < 0.0001, and Treatment; *p* = 0.0343).

**Figure 5 ijms-24-03599-f005:**
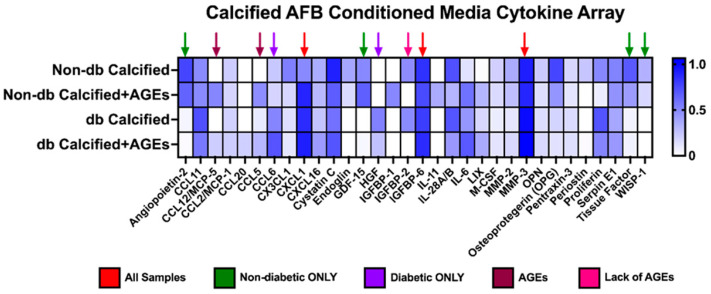
Calcified AFBs-conditioned media from non-diabetic and diabetic cells released different cytokines, chemokines, and proteins. Pooled calcified AFBs-conditioned media (*n* = 3–4) was utilized for a cytokine array kit and manufactured instructions were followed. Densiometric analysis was used for each spot and normalized to the average of the reference spots for each membrane. No statistical analysis was performed.

**Figure 6 ijms-24-03599-f006:**
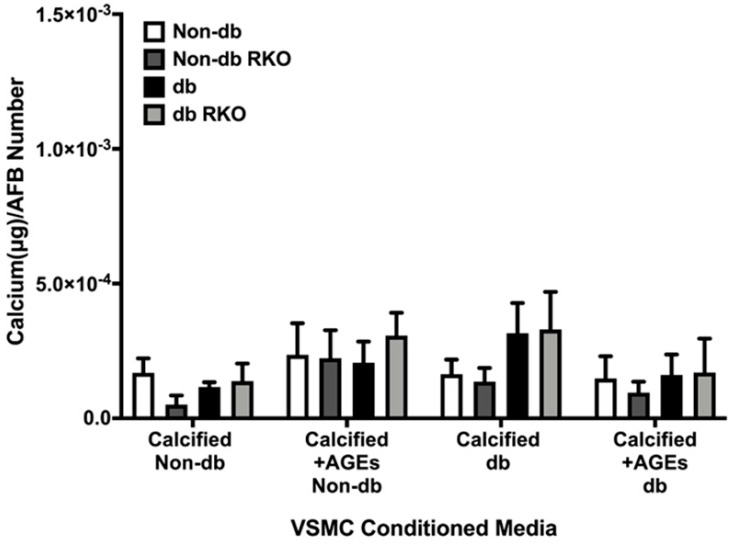
AFBs did not differentially respond to non-diabetic and diabetic VSMCs-conditioned media. Primary AFBs were isolated and treated with VSMCs-conditioned media at a dilution of 1:10 in 2% FBS genotype appropriate DMEM for 7 days. Calcium (μg) was normalized to the number of cells (DAPI) and graphs as mean ± SEM with *n* = 3–6 of independent replicates. Statistical analysis consisted of a two-way ANOVA (*p* = 0.9727).

**Figure 7 ijms-24-03599-f007:**
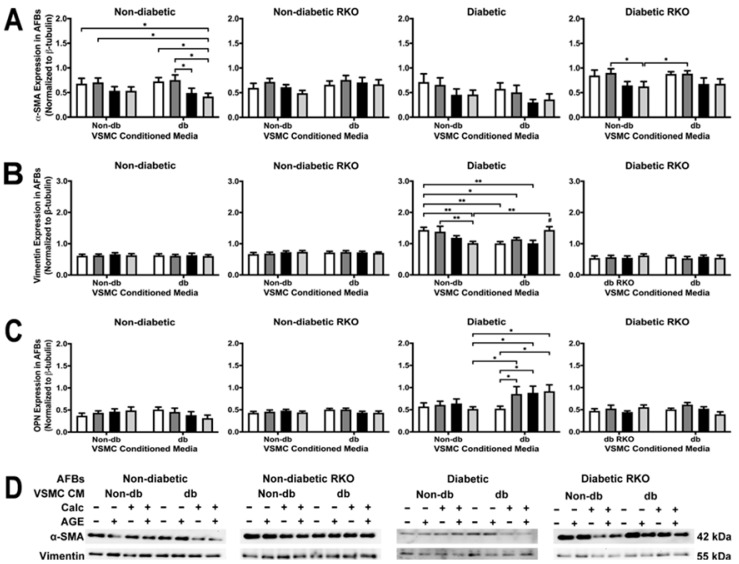
Primary AFBs were isolated and treated with VSMCs-conditioned media at a 1:10 dilution in 2% FBS genotype appropriate DMEM for 7 days. Protein expression was quantified of α-SMA (42 kDa: (**A**)), vimentin (55 kDa: (**B**)) and OPN (32 kDa: (**C**)). All protein levels were normalized to β-tubulin, and (**D**) representative western blot images. Data was graphed as mean ± SEM with *n* = 3–6 of independent replicates. A two-way ANOVA with Fisher’s LSD post hoc was performed (# represents significant to all other groups within treatment group, * *p* < 0.05, ** *p* < 0.01) on each graph individually. All data were analyzed together with a two-way ANOVA to give the following: α-SMA (Interaction; *p* = 0.8482, Genotype; *p* < 0.0001, and Treatment; *p* = 0.0003), Vimentin (Interaction; *p* = 0.0002, Genotype; *p* < 0.0001, and Treatment; *p* = 0.2291), and OPN (Interaction; *p* = 0.0202, Genotype; *p* < 0.0001, and Treatment; *p* = 0.2077).

**Figure 8 ijms-24-03599-f008:**
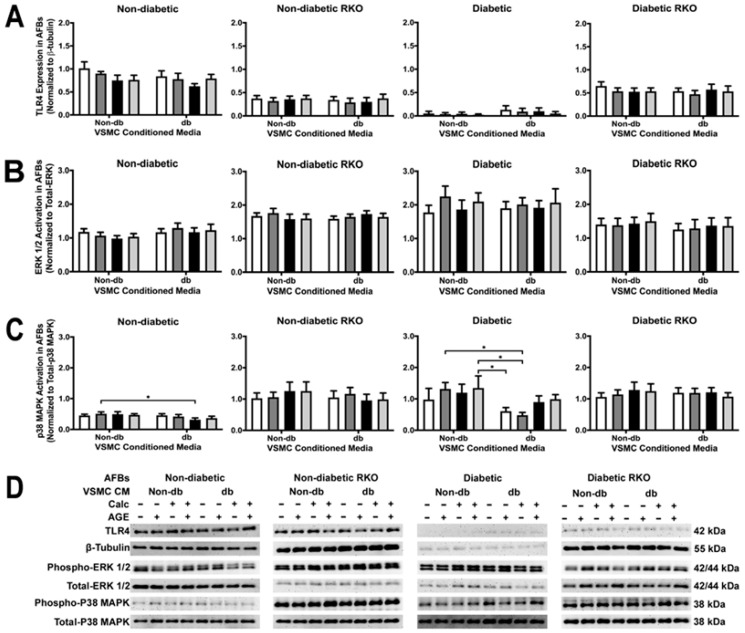
RAGE presence caused a decrease in p38 MAPK activation in AFBs due to diabetic-conditioned media while ERK activation TLR4 expression remained the same. Protein expression was quantified of TLR4 (95 kDa: (**A**)), ERK 1/2 (42/44 kDa: (**B**)) and p38 MAPK (38 kDa: (**C**)). All protein levels were normalized to β-tubulin and (**D**) representative western blot images. Data was and graphed as mean ± SEM with *n* = 3–6 of independent replicates. A two-way ANOVA with Fisher’s LSD post hoc was performed (* *p* < 0.05) on each graph individually. All data were analyzed together with a two-way ANOVA to give the following: TLR4 (Interaction; *p* = 0.9501, Genotype; *p* < 0.0001, and Treatment; *p* = 0.5220), ERK 1/2 (Interaction; *p* = 0.9976, Genotype; *p* < 0.0001, and Treatment; *p* = 0.9408), and p38 MAPK (Interaction; *p* = 0.9292, Genotype; *p* < 0.0001, and Treatment; *p* = 0.2917).

**Figure 9 ijms-24-03599-f009:**
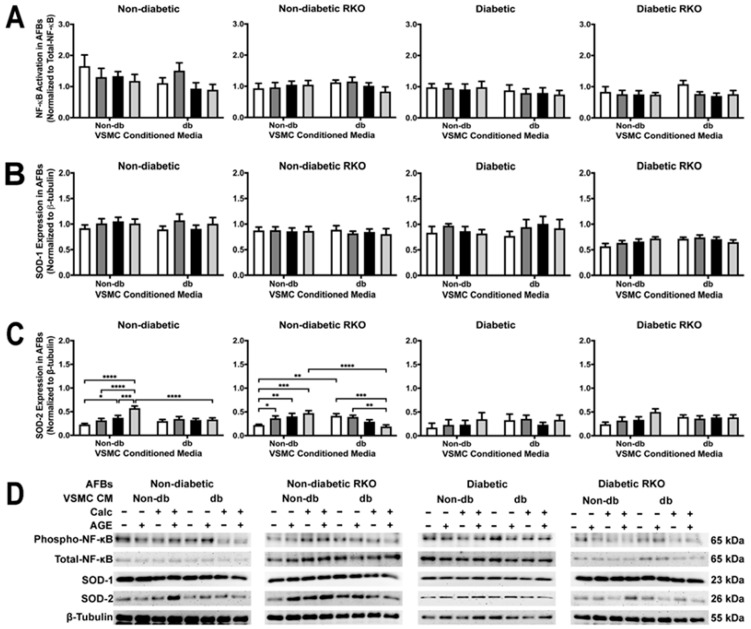
Calcified VSMC-conditioned media caused changes in SOD-2 expression in non-diabetic AFBs. Primary AFBs were isolated and treated with VSMCs-conditioned media at a 1:10 dilution in 2% FBS genotype appropriate DMEM for 7 days. Protein expression was quantified of p65 NF-κB (65 kDa; (**A**)), SOD-1 (23 kDa, (**B**)), and SOD-2 (26 kDa, (**C**)). All protein levels were normalized to β-tubulin (**D**) and graphed as mean ± SEM with *n* = 3–6 of independent replicates. A two-way ANOVA with Fisher’s LSD post hoc was performed (* *p* < 0.05, ** *p* < 0.01, *** *p* < 0.001, **** *p* < 0.0001) on each graph individually. All data were analyzed together with a two-way ANOVA to give the following: NF-κB (Interaction; *p* = 0.8284, Genotype; *p* < 0.0001, and Treatment; *p* = 0.2711), SOD-1 (Interaction; *p* = 0.9545, Genotype; *p* < 0.0001, and Treatment; *p* = 0.8233), and SOD-2 (Interaction; *p* = 0.7592, Genotype; *p* = 0.0508, and Treatment; *p* < 0.0001).

**Figure 10 ijms-24-03599-f010:**
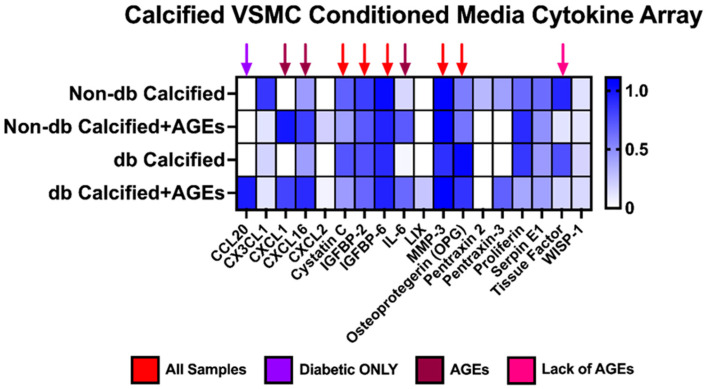
Calcified VSMCs-conditioned media from non-diabetic and diabetic cells released different cytokines, chemokines, and proteins. Pooled calcified VSMCs-conditioned media (*n* = 3–4) was utilized for a cytokine array kit and manufactured instructions were followed. Densiometric analysis was used for each spot and normalized to the average of the reference spots for each membrane. No statistical analysis was performed.

## Data Availability

Not applicable.

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
