# Peer review of "Paracrine Signals in Calcified Conditioned Media Elicited Differential Responses in Primary Aortic Vascular Smooth Muscle Cells and in Adventitial Fibroblasts"

_ijms, 2023, doi:10.3390/ijms24043599_

Round 1
Reviewer 1 Report
General Comments
The manuscript is well-written with broad referencing. The authors investigated “Paracrine Signals in Calcified Conditioned Media Elicited Differential Responses in Primary Aortic Vascular Smooth Muscle Cells and in Adventitial Fibroblast”. The study is interesting and adds to the existing body of knowledge. All the data in the manuscript are well presented. There are a few things that need clarification and revisions. All details and comments are listed below.
Details Comments
1. Page 1, Line 42: vascular smooth muscle cells (VSMCs).
2. Page 2, Line 54: a-SMA and OPN; please write the full name and then abbreviate it.
3. Page 9, Line 158: Figure 7A.
4. Result section: Please replace Figures 2, 3, 4, 7, 8, and 9 with high-quality figures.
5. Page 15, Line 340: VC; please write the full name and then abbreviate it.
6. Page 16, Line 354: Please state the total number of mice used in the current study.
7. Page 16, Line 367-368: Please add citations for the cervical dislocation procedure.
8. Page 19, Line 469: Supplementary Table 1; P-value represents for which group? eg. Diabetic vs Non-diabetic, Diabetic RKO vs Non-diabetic RKO, Diabetic RKO vs non-diabetic. Please add a superscript for the P-value in the table legend.
9. Page 20, Line 472: Please add the full name for each abbreviation in Supplementary Table 2 legend.
Author Response
We would like to thank the reviewers for their time and energy to increase the rigor and relevance of this manuscript. The authors have hopefully addressed the comments raised. Larger figure images have been included at the end of the article. Other formats are available should it be required. Again, thank you for your insightful comments.

Reviewer 2 Report
The manuscript by Amber M. Kennon et al. illustrated that paracrine signals in calcified conditioned media could elicit differential responses in primary aortic vascular smooth muscle cells and in adventitial fibroblasts. The study is well-designed and is of interest to the readers. I have the following suggestions:
1, the authors must revise the abstract. The abstract should be a total of about 200 words maximum.
2, I can hardly read the labelings of the figures 8 and 9. Some other figures also have this problem. The authors must doeble-check and revise.
3, The conditioned media are very important in this study. In the introduction part, the authors must further discuss the defination and the utilization of the conditioned media in relevant studies. Why conditioned media are important? Why did the authors use conditioned media?
Author Response

(The authors gave the same response as above.)

Round 2
Reviewer 2 Report
The authors have revised the manuscript accordingly. It can be considered for publication.